# Analyzing the performance of deep learning splice prediction algorithms

Nathan Fortier*, Gabe Rudy, Andreas Scherer

Research and Development, Golden Helix, Inc., Bozeman, Montana, United States of America

* fortier@goldenhelix.com

## Abstract

SpliceAI is the leading tool for predicting splice-altering variants, but restrictive licensing limits clinical adoption. While open-source implementations have been published with author-reported comparisons, independent benchmarking across diverse datasets is needed to establish equivalence. We compared the original SpliceAI with two open-source implementations (OpenSpliceAI and CI-SpliceAI) and a legacy ensemble baseline across six datasets: a curated set of 1,316 validated variants, 213 variants with splice-assay data, 99,601 variants from the SPiP splicing prediction study, 242 manually curated deep intronic pathogenic variants, and two ClinVar-derived datasets comprising 53,600 intronic variants and 58,064 variants spanning all genomic contexts. The deep learning models were also evaluated against an ensemble of four legacy splice-prediction tools. Across all datasets, the deep learning algorithms outperformed the legacy ensemble. All three deep learning algorithms showed similar performance on the larger datasets dominated by canonical splice site variants (balanced accuracies 0.889-0.977). On the deep intronic benchmark, the original SpliceAI achieved the highest balanced accuracy (0.940), outperforming both CI-SpliceAI (0.890) and OpenSpliceAI (0.841). Critically, optimal thresholds for deep intronic variants were an order of magnitude lower than standard recommendations, indicating that default thresholds would miss the majority of pathogenic deep intronic variants. A correlation analysis showed that CI-SpliceAI maintained balanced concordance across event types, whereas OpenSpliceAI showed stronger correlation for loss events than gain events. Both implementations showed high positional agreement with SpliceAI, with exact splice-site match rates exceeding 90% across event types. Together, these results demonstrate that both open-source reimplementations of SpliceAI successfully reproduce the predictive behavior of the original algorithm across multiple evaluation contexts, while consistently outperforming traditional splice prediction methods. However, performance diverges on deeply intronic variants, and standard score thresholds are poorly calibrated for this variant class regardless of algorithm choice.

**Data availability statement:** The data used in the CI-SpliceAI benchmark analysis are available https://github.com/YStrauch/CI-SpliceAI__Comparison. The Riepe benchmark data are available at https://github.com/cmbi/Benchmarking_splice_prediction_tools. The ClinVar benchmark dataset is available from bioRxiv at https://www.biorxiv.org/content/10.1101/2025.11.20.689501v1.supplementary-material The converted SPiP benchmark dataset is available at https://github.com/nlfortier/Analyzing-the-performance-of-deep-learning-splice-prediction-algorithms/tree/main/SPiP%20Benchmark The Barbosa benchmark data are available in at https://github.com/PedroBarbosa/DeepIntronic_Benchmark.

**Funding:** This research received no external funding. All authors are employees of Golden Helix, Inc., which provided support in the form of salaries for authors Nathan Fortier, Gabe Rudy, and Andreas Scherer, equipment used to perform the analysis, and access to the VarSeq software. The funder did not have any additional role in the study design, data collection and analysis, decision to publish, or preparation of the manuscript. The specific roles of these authors are articulated in the 'author contributions' section.

**Competing interests:** All authors are employees of Golden Helix, Inc., which develops and distributes the VarSeq software containing the Legacy Ensemble splice prediction algorithms evaluated in this study. As employees, all authors receive salaries and standard employment benefits from Golden Helix, Inc. Golden Helix has no commercial relationship with the developers of SpliceAI, OpenSpliceAI, or CI-SpliceAI. The authors hold no relevant patents, consultancy roles, or financial interests in any other products or technologies evaluated in this study. The design, conduct, and reporting of this research were not influenced by commercial considerations. This does not alter our adherence to PLOS ONE policies on sharing data and materials.

## Introduction

Aberrant splicing is a major contributor to human disease, with an estimated 15–30% of all disease-causing variants having the potential to disrupt normal splicing [1]. While splice-altering variants that disrupt the canonical AG and GT splice dinucleotides are easily identified, it is common for non-coding and missense variants lying outside of these dinucleotides to disrupt normal splicing. The determination of whether a given variant will disrupt an existing splice site or introduce a novel splice site has proven to be a computationally difficult problem [2].

Early splice prediction methods attempted to model splice motifs probabilistically. SpliceSiteFinder, one of the first widely adopted tools, used a simple position weight matrix to characterize the nucleotide probabilities surrounding splice sites [3]. GeneSplicer improved upon this by applying Markov models and maximal dependence decomposition to better capture dependencies within the motif [4]. Another commonly used method, MaxEntScan, employed maximum entropy modeling to approximate the distributions underlying true splice sites [5].

Neural network–based approaches soon followed. NNSplice represented the first published attempt to use machine learning for this task, applying a shallow feed-forward neural network with a single hidden layer [6]. The introduction of SpliceAI marked a significant advancement in splice site prediction, leveraging deep convolutional neural networks and large genomic contexts. Using a 10,000 base pair window around each candidate splice site, SpliceAI demonstrated substantially improved accuracy over earlier probabilistic and machine-learning models [7].

Since its publication, SpliceAI has become the leading tool for splice site prediction. While the SpliceAI source code is openly available, its pre-trained model weights and precomputed score datasets are licensed exclusively for academic and non-commercial use, prohibiting their integration into commercial clinical diagnostic pipelines. In response, several authors have developed permissively licensed open-source reimplementations and updates to the original model.

OpenSpliceAI is one such open-source method. This approach is a complete reimplementation and retraining of SpliceAI using the PyTorch machine learning library [8]. Another open-source alternative to SpliceAI is CI-SpliceAI, which offers a reimplementation of the algorithm based on an updated version of the TensorFlow library and a retrained model using a collapsed isoform set representative of all manually annotated constitutive and alternative splice sites from GENCODE [9].

While both OpenSpliceAI and CI-SpliceAI authors reported favorable comparisons to the original SpliceAI, independent validation across diverse datasets is lacking. Furthermore, the degree of score concordance and spatial agreement between implementations has not been systematically quantified.

In this study, we perform a comprehensive comparison of SpliceAI, OpenSpliceAI, CI-SpliceAI, and a voting ensemble of four legacy methods using six benchmark datasets, including curated functional assays, literature-derived variant sets, a large-scale collection of experimentally validated spliceogenic variants, manually curated deep intronic pathogenic variants, and clinical variant classifications from ClinVar. Additionally, we assess the quantitative concordance between open-source

implementations and the original SpliceAI through detailed correlation analysis of predicted splice scores and positions. Our analysis provides empirical guidance for algorithm selection, establishes whether open-source alternatives can serve as viable replacements for the original SpliceAI in research and clinical applications, and examines the extent to which standard score thresholds are appropriate across different variant classes.

## Materials and methods

### Splice prediction tools

This study compares the performance of four different splice prediction tools:

1. **SpliceAI:** the original deep learning splice prediction algorithm trained on GENCODE v24 annotations [7].

2. **OpenSpliceAI:** a PyTorch-based reimplementation of SpliceAI trained on RefSeq MANE v1.3 annotations [8].

3. **CI-SpliceAI:** a SpliceAI reimplementation trained exclusively on manually curated GENCODE HAVANA annotations [9].

4. **Legacy Ensemble:** performs an ensemble vote (N-of-4) across the four established splice prediction algorithms provided by the VarSeq genomic analysis software: MaxEntScan [5], NNSplice [6], GeneSplicer [4], and a position weight matrix algorithm based on SpliceSiteFinder [3].

The inclusion of the Legacy Ensemble enables evaluation of how much additional predictive accuracy the deep learning–based SpliceAI models provide relative to conventional splice-site prediction strategies. For all benchmark analyses, the Legacy Ensemble voting threshold providing the highest balanced accuracy was used for classification. This threshold was $N>1$ for all datasets, with the exception of the deeply intronic variants in the Barbosa dataset and the ABCA4 variants in the Riepe Benchmark dataset, for which the optimal thresholds were $N>0$ and $N>2$ respectively.

For all deep learning algorithms, the following options were used:

- **Masking** was enabled to restrict predictions to biologically meaningful splice alterations, specifically gains of novel splice sites and losses of established splice sites. This avoids reporting spurious events such as gains at positions already annotated as splice sites or losses at positions that do not function as splice sites in the reference annotation.

- **Maximum distance** between the variant and a gained or lost splice site was set to 500 bp, consistent with the default configuration of the Broad Institute SpliceAI Lookup web interface (spliceailookup.broadinstitute.org).

**Technical overview of deep learning methods.** All three deep learning algorithms evaluated in this study share a common architectural framework which utilizes deep residual neural networks to predict splice sites directly from the genomic DNA sequence [7]. For each position in a genomic sequence, the models output probabilities indicating whether that position functions as a splice donor site, splice acceptor site, or neither. By comparing predictions generated from reference and variant sequences, these methods can detect both disruption of canonical splice sites and activation of novel cryptic splice sites.

The core architecture consists of 32 dilated convolutional layers, enabling the network to capture sequence features across large genomic distances. This design is critical for splice prediction, as regulatory elements influencing splice donor and acceptor activation may be separated by tens of thousands of nucleotides. Dilated convolutions allow integration of both proximal and distal sequence context without requiring an impractically large number of parameters.

The original SpliceAI was implemented using TensorFlow/Keras [7] and trained on GENCODE v24GRCh37 [10] pre-mRNA transcripts from a subset of human chromosomes, with additional splice junctions derived from GTEx RNA-sequencing data. While incorporation of GTEx data expanded the diversity of observed splice events, the exact processing and filtering criteria for these junctions were not fully documented, limiting independent assessment of their quality.

OpenSpliceAI is an open-source reimplementation of SpliceAI that preserves the original network architecture while adopting the PyTorch framework [8]. This results in improved computational efficiency but produces models that are not identical to the original SpliceAI due to a variety of factors including weight initialization, data shuffling, and batch normalization behavior. Additionally, OpenSpliceAI utilizes the AdamW optimizer, which incorporates exponential moving averages of gradient moments and introduces stochastic elements into the adaptive learning process. OpenSpliceAI was trained using protein-coding transcripts from the RefSeq MANE v1.3 dataset, which provides a curated, one-to-one mapping between RefSeq and Ensembl/GENCODE annotations on GRCh38.

CI-SpliceAI retains the same neural network architecture but modifies the training strategy to address potential limitations of earlier implementations [9]. Specifically, CI-SpliceAI excludes RNA-seq-derived splice junctions and instead relies exclusively on a collapsed isoform set of manually curated splice sites from GENCODE HAVANA annotations. This approach prioritizes high-confidence constitutive and alternative splice sites while avoiding potential mismatches between genomic sequence input and tissue/individual-specific splicing events inferred from transcriptomic data. In contrast to the original SpliceAI, CI-SpliceAI was trained using data from all human chromosomes.

## Performance benchmarking methodology

The performance of these algorithms was evaluated across three different datasets. For each dataset, the sensitivity, specificity, F1 score, and balanced accuracy were computed as follows:

$$\text{Sensitivity} = \frac{TP}{TP + FN} \tag{1}$$

$$\text{Specificity} = \frac{TN}{TN + FP} \tag{2}$$

$$\text{F1 score} = \frac{2 \times TP}{2 \times TP + FP + FN} \tag{3}$$

$$\text{Balanced Accuracy} = \frac{Sensitivity + Specificity}{2} \tag{4}$$

To determine if the observed differences in overall accuracy were statistically significant, a two-proportion Z-test was performed to compare the performance of the open-source models (CI-SpliceAI and OpenSpliceAI) against the original SpliceAI algorithm with statistical significance assessed using a significance threshold of $P\text{-value} < 0.05$.

**CI-SpliceAI benchmark dataset.** The first dataset comprises 1,316 unique variants aligned to the GRCh37 reference sequence with functionally validated splicing impacts sourced from the literature. This dataset was published by the original CI-SpliceAI authors and is balanced between splice-altering and non-altering variants, as well as between strand orientations, reducing potential biases [9] (see S1 File). Within this dataset, 25.1% of the variants alter the canonical splice site dinucleotides.

A binary classification approach was adopted to assess each algorithm's ability to distinguish between splice-altering and non-splice-altering variants.

• Benchmark variants labeled as alternative acceptor, alternative donor, multiple, retention, or skip were classified as Splice-Altering (Positive).

• Variants labeled normal were classified as Not Splice-Altering (Negative).

For both models, the maximum of the four scores (Acceptor Gain, Acceptor Loss, Donor Gain, Donor Loss) was used. A variant was classified as Splice-Altering (Positive) if the maximum score was > 0.5, corresponding to the recommended threshold for pathogenic classification by SpliceAI [7].

**Riepe benchmark dataset.** The second analysis evaluated performance on the functional splice assay dataset published by Riepe *et al.*, consisting of 152 ABCA4 and 61 MYBPC3 variants [11] (see S2 File). Notably, 81 of the ABCA4 variants are deeply intronic and none of the variants in this dataset alter the canonical splice site dinucleotides. Functional validation data from LOVD, ClinVar, and ExAC provided precise ground-truth labels distinguishing between specific splice site alterations (acceptor/donor gain or loss).

For this benchmark, the most relevant SpliceAI score was selected based on each variant's effect type, and score thresholds were optimized per gene and algorithm combination to yield the highest possible accuracy, thus representing an upper-bound estimate of model performance on functional data.

**SPiP benchmark dataset.** To evaluate algorithm performance on a large, diverse collection of variants with experimentally validated splicing outcomes, we used the curated dataset published by Leman *et al.* in their development of the Splicing Prediction Pipeline (SPiP) [12] (see S4 File). This dataset includes 4,616 variants from 227 genes involved in 161 clinical signs and syndromes, of which 1,924 (41.68%) are proven spliceogenic variants supported by *in vitro* RNA studies including RT-PCR, Sanger sequencing of RT-PCR products, and/or minigene splicing assays. The spliceogenic variants encompass the full spectrum of splicing impact including exon skipping, splice site shift, pseudoexon creation, and full intronic retention. Only 0.6% of the variants in this dataset alter the canonical splice site dinucleotides.

In addition to the 4,616 experimentally characterized variants, the original authors supplemented the dataset with 95,000 control variants with minor allele frequency > 5%. This yielded a comprehensive collection of 99,616 variants.

Variants were classified as splice-altering (positive) or non-splice-altering (negative) according to the binary `class_splice` field provided by the original authors, where a value of 1 indicates a spliceogenic variant and 0 indicates a non-spliceogenic variant. This classification encompasses both the experimentally characterized variants and the common variant controls.

The original dataset was provided in HGVS notation. We converted the variants to VCF format aligned to the GRCh38 reference assembly. During conversion, 15 variants were excluded due to improperly formatted HGVS notation or complex variant types (large deletions, indels) that could not be unambiguously resolved. The specific variants excluded are listed in the Supporting Information.

After exclusion of these variants, the final dataset consisted of 99,601 variants for evaluation.

**Barbosa deep intronic benchmark dataset.** To specifically assess performance on deeply intronic variants, we ran the algorithms on the manually curated dataset of disease-causing deep intronic variants published by Barbosa *et al.* [13] (see S5 File). This dataset contains 242 pathogenic intronic variants, each located more than 10 base pairs from the nearest annotated splice site and supported by experimental evidence of aberrant splicing (e.g., RT-PCR, cDNA sequencing, RNA-seq, or minigene assays). The majority of these variants (194 of 242) cause pseudoexon activation, with the remainder primarily leading to partial intron retention.

A matched negative set of 242 common variants (allele frequency > 1%) was drawn from gnomAD v2.1. Negative variants were selected from 500 bp windows surrounding the positive variants, filtered to retain only intronic variants occurring in the same set of genes, and confirmed to be absent from ClinVar.

As with the other benchmarks, the maximum of the four delta scores was used for classification. However, because this dataset is restricted to deeply intronic variants, we determined optimal score thresholds by maximizing balanced accuracy on this dataset rather than relying on the standard recommended threshold of 0.5. This approach was motivated by the finding of Barbosa *et al.* that deeply intronic variants produce substantially weaker prediction scores than canonical splice site variants, with the optimal threshold for SpliceAI determined by the authors to be 0.05, which is an order of magnitude lower than the standard recommended threshold.

**Barbosa ClinVar benchmark dataset.** To further evaluate algorithm performance on intronic variants, we used the ClinVar benchmark dataset published by Barbosa *et al.* [13] (see S5 File). This dataset consists of 53,600 GRCh37-aligned intronic single-nucleotide variants from ClinVar (release date: 2022−04), including both pathogenic/likely pathogenic and benign/likely benign variants, each with at least one-star review confidence.

Because this dataset was not filtered by distance to the nearest splice site, it reflects the composition of ClinVar itself: more than 90% of pathogenic intronic variants occur at canonical splice site positions, and more than 95% fall within 10 nucleotides of an exon-intron boundary [13]. Furthermore, 22.8% of the variants in this dataset alter the canonical splice site dinucleotides. This enrichment for near-splice-site variants provides a useful contrast to the deeply intronic benchmark described above.

Positive and negative labels were assigned based on ClinVar clinical significance: variants classified as Pathogenic or Likely Pathogenic were designated as splice-altering (positive), and those classified as Benign or Likely Benign were designated as non-splice-altering (negative). For each algorithm, the maximum of the four delta scores (Acceptor Gain, Acceptor Loss, Donor Gain, Donor Loss) was used, and variants were classified as splice-altering if the maximum score exceeded the recommended threshold of 0.5.

**Extended ClinVar benchmark dataset.** The final benchmark dataset was constructed from the ClinVar Assessments database (release date: 2025−10) [14] (see S3 File). All GRCh38-aligned variants were imported into the VarSeq software and subjected to an initial pre-filtering step in which we removed variants for which none of the four Legacy Ensemble splice prediction algorithms indicated either a gained or disrupted splice site.

The purpose of this pre-filtering step was to construct a benchmark that includes benign variants with computational evidence of splicing impact. Without pre-filtering, the negative set would be dominated by variants with no computational evidence of splice relevance, artificially inflating specificity estimates for all algorithms. By retaining only benign variants for which at least one Legacy Ensemble algorithm predicted a splice effect, we ensure that the negative set includes variants that could plausibly be mistaken as splice-altering by computational tools. An alternative approach of restricting the dataset to synonymous or intronic variants was considered but not adopted, as this would exclude missense variants with documented splicing effects.

We note that this filtering strategy does introduce a potential bias. Specifically, variants detectable only by deep learning models but not by any Legacy Ensemble algorithm are systematically excluded, which may conservatively underestimate the specificity advantage of deep learning approaches. The impact of this limitation is partially mitigated by the inclusion of the Barbosa deep intronic benchmark, which evaluates algorithm performance on deeply intronic variants without any pre-filtering based on legacy tool predictions.

Following pre-filtering, we applied two independent classification filter chains to assign variants to positive and negative benchmark sets:

1. **Splice-Altering (Positive Set):** Variants classified by ClinVar as Pathogenic and whose submitted interpretation or summary text contained the term "splice." These variants are presumed to have experimentally or clinically supported splice-altering effects.

2. **Non–Splice-Altering (Negative Set):** Variants classified by ClinVar as Benign. These variants are assumed not to disrupt normal splicing.

Variants meeting the first criterion were designated as splice-altering, whereas those meeting the second were designated as non-disruptive. This procedure resulted in a combined benchmark dataset of 58,064 variants.

The dataset was intentionally constructed using fully automated criteria to enable evaluation at a scale that would be impractical with manual variant curation. However, the composition of the positive set reflects known biases in the ClinVar Assessments databases. Of the variants in this dataset, 17.4% alter the canonical splice site dinucleotides. Additionally, ClinVar contains an overrepresentation of pathogenic variants affecting the immediately flanking consensus motif, which are relatively straightforward for prediction algorithms to identify [9,13]. Consequently, performance on this dataset

does not provide a measure of algorithm sensitivity for non-canonical or deeply intronic splice variants, which are better assessed using the Barbosa deep intronic benchmark described above.

An additional limitation is that the negative set does not include pathogenic variants with confidently established non-splicing mechanisms. At ClinVar scale, reliably identifying such variants without manual curation is not feasible, as absence of splicing annotation does not imply absence of splicing effects. Furthermore, unlike the Barbosa ClinVar dataset [13], which applied review confidence filtering, our automated approach prioritizes breadth over curation stringency. Despite these limitations, this dataset provides a comprehensive evaluation of algorithm performance across diverse genomic contexts and complements the more targeted benchmarks described above.

**Correlation analysis.** To quantify the concordance between the scores generated by the open-source algorithms (CI-SpliceAI and OpenSpliceAI) and those from the original SpliceAI algorithm, a correlation analysis was performed. Variants were sampled from the precomputed SpliceAI scores provided by Illumina for the GRCh37 reference genome assembly [7].

Prior to random sampling, the dataset was filtered to exclude variants with minimal evidence of splicing disruption by removing any scores for which all SpliceAI delta scores were below 0.1. A final set of 100,000 variants was randomly sampled from this filtered subset, comprising 50,000 from the masked indel file and 50,000 from the masked SNV (single nucleotide variant) file.

To enable a paired comparison, both the OpenSpliceAI and CI-SpliceAI algorithms were executed to obtain new predictions for the 100,000 variants. Masked delta scores were computed for both algorithms using a maximum distance of 50 bp between the variant and the candidate splice site, thereby matching the parameters used to generate the original precomputed SpliceAI scores.

Four splice disruption categories were evaluated:

1. DS_AG: Delta Score Acceptor Gain

2. DS_AL: Delta Score Acceptor Loss

3. DS_DG: Delta Score Donor Gain

4. DS_DL: Delta Score Donor Loss

Each score represents the probability that the variant is splice-altering. The authors recommend a threshold of 0.2 for high recall, 0.5 for balanced accuracy, and 0.8 for high precision. For each score, the following correlation statistics were computed.

- **Spearman's rank correlation coefficient ($\rho$):** A non-parametric measure of monotonic association between CI-SpliceAI and SpliceAI scores. Spearman correlation was chosen over Pearson correlation because splice scores are bounded (0–1) and the relative ranking of variants is more clinically relevant than absolute numerical agreement.

- **Mean Absolute Difference (MAD):** The average absolute difference between paired scores providing an interpretable measure of typical score divergence.

For variants where either tool predicted a delta score ≥ 0.2 (the high-recall threshold recommended by the SpliceAI authors), we assessed whether the predicted delta positions matched exactly. The delta position (DP) indicates the distance (in base pairs) from the variant to the predicted affected splice site.

For each score type we computed the following:

- **Exact Match Rate:** The percentage of high-confidence predictions for which the delta position matches that reported by SpliceAI.

- **95% Confidence Intervals:** Wilson score intervals for the match rate, accounting for binomial uncertainty.

# Results

## CI-SpliceAI benchmark results

When compared on the CI-SpliceAI benchmark data, all three deep learning algorithms demonstrated highly similar, strong performance, with overall balanced accuracies exceeding 0.89 and F1 scores exceeding 0.88 (Table 1). These results seem to confirm that both algorithms successfully replicate the general predictive capability of the original SpliceAI model.

Although CI-SpliceAI achieved the highest balanced accuracy (and F1 score) on this dataset, neither CI-SpliceAI (two-proportion Z-test, *P*-value = 0.460) nor OpenSpliceAI (*P*-value = 0.844) demonstrated a statistically significant improvement in accuracy over the original SpliceAI model. In contrast, the Legacy Ensemble performed significantly worse than all SpliceAI-based methods, with a balanced accuracy of 0.785 (*P*-value < 0.001).

## Riepe benchmark results

Unlike the first analysis, this dataset revealed a more pronounced performance divergence (Table 2) among the models.

While the original SpliceAI achieved the highest balanced accuracy (and F1 score) at 0.839 and 0.836 respectively, CI-SpliceAI showed no statistically significant difference in accuracy (*P*-value = 0.219). However, OpenSpliceAI demonstrated a significantly lower balanced accuracy than SpliceAI at 0.746 (*P*-value = 0.019). All three algorithms significantly outperformed the Legacy Ensemble approach (*P*-value < 0.0001), which achieved a balanced accuracy of just 0.659.

## SPiP benchmark results

The SPiP benchmark dataset presents a substantially different evaluation context from the other benchmarks due to its extreme class imbalance, with spliceogenic variants constituting approximately 2% of the total dataset [12]. Performance metrics should be interpreted accordingly (Table 3).

All three deep learning algorithms achieved near-identical specificity and similar sensitivity. While SpliceAI achieved a marginally better balanced accuracy of 0.803 compared to 0.798 for CI-SpliceAI and 0.787 for OpenSpliceAI, neither difference reached statistical significance (*P-value* = 0.250 and 0.248, respectively).

**Table 1. Performance metrics for CI-SpliceAI benchmark.**

| Method | Sensitivity | Specificity | F1 | Balanced Accuracy |
|---|---|---|---|---|
| SpliceAI | 0.840 | 0.956 | 0.891 | 0.898 |
| CI-SpliceAI | 0.837 | 0.976 | 0.899 | 0.907 |
| OpenSpliceAI | 0.816 | 0.975 | 0.886 | 0.895 |
| Legacy Ensemble | 0.730 | 0.840 | 0.772 | 0.785 |

**Table 2. Performance metrics for Riepe benchmark.**

| Method | Sensitivity | Specificity | F1 | Balanced Accuracy |
|---|---|---|---|---|
| SpliceAI | 0.790 | 0.883 | 0.839 | 0.836 |
| CI-SpliceAI | 0.706 | 0.883 | 0.785 | 0.794 |
| OpenSpliceAI | 0.672 | 0.819 | 0.741 | 0.746 |
| Legacy Ensemble | 0.500 | 0.817 | 0.609 | 0.659 |

**Table 3. Performance metrics for SPiP benchmark dataset.**

| Method | Sensitivity | Specificity | F1 | Balanced Accuracy |
|---|---|---|---|---|
| SpliceAI | 0.607 | 0.999 | 0.739 | 0.803 |
| CI-SpliceAI | 0.596 | 0.999 | 0.725 | 0.798 |
| OpenSpliceAI | 0.595 | 0.999 | 0.724 | 0.797 |
| Legacy Ensemble | 0.547 | 0.956 | 0.289 | 0.752 |

The moderate sensitivity observed across all three algorithms at the standard 0.5 threshold is consistent with the composition of this dataset, which includes exonic splicing regulatory element disruptions and deep intronic variants that produce weaker prediction signals.

The Legacy Ensemble exhibited the lowest balanced accuracy (0.752) and a markedly lower F1 score (0.289) compared to the deep learning methods. This sharp discrepancy between F1 and balanced accuracy is driven by a substantially higher false-positive rate: the Legacy Ensemble produced 4,247 false positives compared to 70–94 for the deep learning algorithms, resulting in a specificity of 0.956 versus 0.999. In the context of whole-genome variant filtering, where the vast majority of variants are non-spliceogenic, this difference in specificity translates to a substantially higher burden of false positives for downstream interpretation.

## Barbosa deep intronic benchmark results

In contrast to the previous benchmarks, performance on the Barbosa deep intronic dataset revealed notable differences between algorithms when evaluated using optimized thresholds (Table 4). Optimal thresholds were determined by maximizing balanced accuracy, resulting in thresholds of 0.04 for SpliceAI and 0.01 for both CI-SpliceAI and OpenSpliceAI. For the Legacy Ensemble, the optimal threshold was $N > 0$.

SpliceAI achieved the highest balanced accuracy (0.940) and sensitivity (0.930), statistically significantly outperforming both CI-SpliceAI ($P\text{-value} < 0.001$) and OpenSpliceAI ($P\text{-value} < 0.001$). OpenSpliceAI exhibited the lowest sensitivity among all algorithms (0.702) but the highest specificity (0.979). The Legacy Ensemble achieved a balanced accuracy of 0.785, performing below all three deep learning methods.

These results suggest that both open-source reimplementations underperform the original SpliceAI on deeply intronic variants, indicating that the differences in training data between these models disproportionately affect the detection of the subtler signals produced by deep intronic variants.

## Barbosa ClinVar benchmark results

Performance on the Barbosa ClinVar benchmark dataset was consistent with the results observed on the CI-SpliceAI benchmark, with all three deep learning algorithms achieving high overall accuracy (Table 5).

**Table 4. Performance metrics for Barbosa deep intronic benchmark using optimized thresholds. Optimal thresholds were determined by maximizing balanced accuracy on this dataset.**

| Method | Sensitivity | Specificity | F1 | Balanced Accuracy |
|---|---|---|---|---|
| SpliceAI | 0.930 | 0.950 | 0.939 | 0.940 |
| CI-SpliceAI | 0.851 | 0.930 | 0.886 | 0.890 |
| OpenSpliceAI | 0.702 | 0.979 | 0.815 | 0.841 |
| Legacy Ensemble | 0.752 | 0.818 | 0.778 | 0.785 |

**Table 5. Performance metrics for Barbosa ClinVar benchmark.**

| Method | Sensitivity | Specificity | F1 | Balanced Accuracy |
|---|---|---|---|---|
| SpliceAI | 0.948 | 0.997 | 0.969 | 0.973 |
| CI-SpliceAI | 0.958 | 0.997 | 0.974 | 0.977 |
| OpenSpliceAI | 0.932 | 0.998 | 0.961 | 0.965 |
| Legacy Ensemble | 0.926 | 0.959 | 0.902 | 0.942 |

CI-SpliceAI achieved the highest balanced accuracy at 0.977, representing a statistically significant improvement over the original SpliceAI algorithm (*P-value* < 0.001). OpenSpliceAI, by contrast, was slightly less accurate than SpliceAI with a balanced accuracy of 0.965 (*P-value* < 0.001). The Legacy Ensemble again demonstrated the lowest balanced accuracy (0.942) and F1 score (0.902) among all methods, driven primarily by a higher false-positive rate as reflected in its lower specificity (0.959) compared to the deep learning algorithms, which all exceeded 0.997.

## Extended ClinVar benchmark results

Performance on the Extended ClinVar benchmark dataset was consistent with the results observed on the Barbosa ClinVar benchmark, with all three deep learning algorithms achieving high overall accuracy (Table 6).

Specifically, CI-SpliceAI and OpenSpliceAI achieved the highest balanced accuracy at 0.895, demonstrating a small, but statistically significant, improvement when compared to the original SpliceAI algorithm (*P-value* < 0.001). Conversely, the Legacy Ensemble performed significantly worse than the other models, yielding the lowest balanced accuracy at 0.809 (*P-value* < 0.001). Notably, the Legacy Ensemble's F1 score (0.684) was substantially lower than its balanced accuracy. This discrepancy indicates a comparatively high number of false positives as evidenced by its lower specificity (0.787) relative to the deep learning methods, which all demonstrated specificities exceeding 0.987.

Notably, all three deep learning algorithms demonstrated lower sensitivity on this dataset than on the Barbosa ClinVar benchmark. This difference is expected given that the pre-filtering strategy employed in the Extended ClinVar benchmark retained only variants for which at least one Legacy Ensemble algorithm predicted splicing impact. As a result, benign variants in this dataset are more likely to possess sequence features that resemble true splice signals, making them harder to correctly classify and consequently depressing sensitivity estimates relative to the Barbosa ClinVar benchmark, which did not pre-filter based on computational evidence.

## Correlation analysis results

**Delta score correlation.** Both open-source models demonstrated strong correlation with the original SpliceAI scores, as measured by Spearman's rank correlation coefficient $\rho$. The degree of correlation varied by the type of splice event (Acceptor/Donor Loss/Gain), as shown in Table 7.

**Table 6. Performance metrics for ClinVar benchmark.**

| Method | Sensitivity | Specificity | F1 | Balanced Accuracy |
|---|---|---|---|---|
| SpliceAI | 0.791 | 0.987 | 0.865 | 0.889 |
| CI-SpliceAI | 0.800 | 0.991 | 0.876 | 0.895 |
| OpenSpliceAI | 0.799 | 0.992 | 0.877 | 0.895 |
| Legacy Ensemble | 0.832 | 0.787 | 0.684 | 0.809 |

**Table 7. Spearman's Rank Correlation ($\rho$) and Mean Absolute Difference (MAD) on Delta Score.**

| Score | CI-SpliceAI | | OpenSpliceAI | |
|---|---|---|---|---|
| | $\rho$ | MAD | $\rho$ | MAD |
| DS_AG | 0.786 | 0.070 | 0.668 | 0.077 |
| DS_AL | 0.864 | 0.029 | 0.924 | 0.024 |
| DS_DG | 0.808 | 0.069 | 0.677 | 0.079 |
| DS_DL | 0.883 | 0.025 | 0.940 | 0.022 |

CI-SpliceAI scores exhibited a strong correlation with original SpliceAI scores, ranging from $\rho$ = 0.786 (DS_AG) to $\rho$ = 0.883 (DS_DL). The mean absolute difference (MAD) was low across all categories, ranging from 0.025 to 0.07.

OpenSpliceAI also demonstrated strong correlations, with particularly high values for loss-of-function events ($\rho$ = 0.924 for DS_AL and $\rho$ = 0.94 for DS_DL). However, its correlation was noticeably lower for gain-of-function events ($\rho$ = 0.668 for DS_AG and $\rho$ = 0.677 for DS_DG) compared to CI-SpliceAI.

**Delta position agreement.** Analysis of the predicted splice site locations for variants with evidence of splicing disruption (delta score ≥ 0.2) showed strong agreement between the open-source models and the original SpliceAI algorithm, as indicated by the high exact match rate (EMR) across splice disruption types (Table 8).

Both models achieved an exact match rate of over 90% for all four splice categories, demonstrating that when a variant is predicted to disrupt splicing, the location of the affected site is nearly always consistent with the original SpliceAI algorithm. The exact match rate was consistently higher for OpenSpliceAI, ranging from 92.5% for DS_AG to its highest at 98.5% for DS_DL.

## Discussion

### Benchmark analysis

These results indicate that both CI-SpliceAI and OpenSpliceAI successfully replicate the core predictive capabilities of SpliceAI, though with notable differences in performance characteristics across different evaluation contexts. Across the four benchmarks dominated by canonical and near-splice-site variants, all three deep learning algorithms achieved high balanced accuracies, generally exceeding 0.89 on the larger datasets with no statistically significant differences in most comparisons. Similarly, on the SPiP benchmark, which evaluates performance under realistic class imbalance, the three algorithms were statistically indistinguishable in balanced accuracy.

**Table 8. Exact Match Rate for Delta Position.**

| Site Type | CI-SpliceAI | OpenSpliceAI |
|---|---|---|
| | Exact Match Rate | Exact Match Rate |
| AG | 90.6% | 92.5% |
| | (CI: 90.2–91.0%) | (CI: 92.1–92.9%) |
| AL | 96.1% | 97.6% |
| | (CI: 95.7–96.4%) | (CI: 97.3–97.8%) |
| DG | 92.5% | 95.5% |
| | (CI: 92.2–92.9%) | (CI: 95.2–95.8%) |
| DL | 97.5% | 98.5% |
| | (CI: 97.2–97.8%) | (CI: 98.2–98.7%) |

While performance was largely equivalent on these datasets, two benchmarks revealed more substantial divergences. On the Riepe dataset of functionally validated splice assays from ABCA4 and MYBPC3, the original SpliceAI achieved the highest balanced accuracy at 0.836, with OpenSpliceAI demonstrating significantly inferior performance at 0.746 ($P$-value = 0.019).

To assess whether this divergence reflects training data coverage differences, we compared the GENCODE v24 annotation used to train SpliceAI with the RefSeq MANE Select annotation used to train OpenSpliceAI at both loci. Two findings are noteworthy. First, ABCA4 resides on a held-out test chromosome under the data split applied by both models, so neither was trained on ABCA4 splice sites; any performance difference at this locus reflects generalisation capacity rather than training coverage. Second, for MYBPC3, GENCODE v24 contributed three protein-coding isoforms encompassing 36 unique introns, compared to the single MANE Select transcript (NM_000256.3) with 34 introns, yielding two introns exclusive to the GENCODE annotation. However, none of the 42 Riepe intronic and splice-region variants in the MYBPC3 dataset fell within these GENCODE-exclusive introns; all mapped to introns present identically in both annotations. These findings indicate that annotation coverage at the specific intronic positions tested is equivalent between models, ruling out training data coverage as an explanation for the observed performance gap. Notably, the Riepe dataset is enriched for deep intronic variants in ABCA4, suggesting that the inferior performance of OpenSpliceAI at this locus may reflect a more general difficulty with deep intronic variant detection rather than a gene-specific effect.

This interpretation is supported by the Barbosa deep intronic benchmark, which provides evidence of the same pattern. SpliceAI achieved a balanced accuracy of 0.940, significantly outperforming both CI-SpliceAI (0.890) and OpenSpliceAI (0.841), despite all three models performing comparably on both ClinVar benchmarks and the CI-SpliceAI benchmark. The convergence of this finding across two independent benchmarks suggests that differences in training data or procedures between the original SpliceAI and its open-source reimplementations disproportionately affect the detection of the subtler splicing signals produced by deep intronic variants.

The SPiP benchmark provided additional context for the interpretation of these findings. On this dataset, all three deep learning algorithms achieved moderate sensitivity (0.595-0.607) at the standard 0.5 threshold. The lack of statistically significant differences between the three deep learning models on this dataset suggests that their performance divergences emerge primarily on specific variant subclasses rather than in aggregate.

## Score correlation and position agreement

The correlation analysis provides additional nuance to our understanding of algorithm concordance. CI-SpliceAI demonstrated relatively balanced correlation across all splice event types, with Spearman's $\rho$ values ranging from 0.786 to 0.883. In contrast, OpenSpliceAI showed a striking asymmetry, with very high correlation for loss events ($\rho$ = 0.924 for DS_AL and $\rho$ = 0.940 for DS_DL) but substantially lower correlation for gain events ($\rho$ = 0.668 for DS_AG and $\rho$ = 0.677 for DS_DG).

This asymmetry suggests that OpenSpliceAI may have learned different feature representations for splice site gains compared to the original SpliceAI, despite achieving similar balanced accuracy. The consistently lower mean absolute difference values for loss events across both algorithms (0.022-0.029) compared to gain events (0.069-0.079) indicates that predicting the disruption of existing splice sites is inherently more consistent across implementations than predicting the activation of cryptic splice sites.

Despite these differences in score distributions, both open-source implementations demonstrated remarkably high agreement with SpliceAI in predicting the locations of affected splice sites. Exact match rates exceeded 90% for all splice event types, reaching as high as 98.5% for donor loss events with OpenSpliceAI. This high spatial concordance is clinically significant, as accurately localizing the affected splice site is often as important as detecting its presence for understanding the disease mechanisms.

## Limitations of sequence-based predictions

All three deep learning algorithms substantially outperformed the Legacy Ensemble approach across all six benchmarks. The Legacy Ensemble achieved balanced accuracies ranging from 0.659 to 0.942, compared to 0.746 to 0.977 for the deep learning methods. The performance gap was most pronounced on the SPiP benchmark, where the Legacy Ensemble's F1 score (0.289) was dramatically lower than those of the deep learning algorithms (0.724-0.739), driven by a false-positive rate roughly 50 times higher. This performance gap demonstrates the transformative impact of deep learning on computational splice prediction.

However, none of the algorithms in our study achieved a balanced accuracy greater than 0.977, and performance on datasets enriched for non-canonical variants was substantially lower. This performance ceiling reflects a fundamental limitation of sequence-based prediction approaches. Because alternative splicing is influenced not only by the underlying RNA sequence but also by the multiple regulatory interactions of various molecular factors within the cell [15], any splice site prediction algorithm based strictly on the DNA sequence will face an asymptotic limit to its maximum achievable accuracy. The SPiP benchmark results illustrate this point: even with optimized thresholds, deep learning methods detected only approximately 60% of splice altering variants.

## Considerations for algorithm selection

Based on these findings, CI-SpliceAI appears to be the strongest open-source alternative for most applications. It demonstrated consistent performance across five of the six benchmarks with no statistically significant decrease in accuracy compared to the original SpliceAI, while exhibiting more balanced correlation patterns across different splice event types compared to OpenSpliceAI.

OpenSpliceAI remains a viable alternative, particularly for applications that prioritize the detection of splice site losses, where it demonstrates the strongest correlation with the original algorithm. However, its relatively weaker performance on both the Riepe functional validation dataset and the Barbosa deep intronic benchmark, suggests caution is warranted for applications requiring the highest possible accuracy for non-canonical or deep intronic variants.

For applications specifically targeting deep intronic variant interpretation, the original SpliceAI demonstrated an advantage over both open-source alternatives. Critically, practitioners working with deep intronic variants should adopt substantially lower score thresholds than the standard recommendations, regardless of which algorithm is used.

For organizations already using the original SpliceAI with access to its precomputed score databases, there is little motivation to adopt an alternative implementation on the basis of accuracy alone. However, commercial clinical laboratories face several practical considerations that extend beyond predictive performance when selecting a splice prediction solution.

First, licensing constraints are a critical barrier. The original SpliceAI model weights and precomputed scores are restricted to academic, non-commercial use, preventing their integration into validated clinical pipelines. This alone necessitates alternative implementations for commercial laboratories.

Second, computational throughput must be considered. Given the volume of variants processed in clinical settings, it is generally infeasible to run deep learning models on every variant in real time. Instead, laboratories must utilize precomputed genome-wide scores for efficient prediction. At present, no official precomputed scores exist for either CI-SpliceAI or OpenSpliceAI, limiting their immediate clinical utility despite the advantage provided by their permissive licensing.

Finally, there are significant limitations associated with the existing SpliceAI precomputed scores. The widely used Illumina-provided datasets were generated only for the GRCh37 reference genome, and the available GRCh38 scores were derived through an error-prone liftover process rather than native computation. In addition, these precomputed scores rely on outdated transcript annotations that no longer reflect current gene models, particularly for genes with extensive alternative splicing.

To address these gaps, new genome-wide splice prediction scores for these alternative methods must be computed natively for both GRCh37 and GRCh38 using up-to-date transcript definitions. Such resources are essential for enabling the adoption of these open-source solutions in high-throughput clinical environments.

## Study limitations

Several limitations of this study should be acknowledged. First, the Riepe benchmark dataset, while consisting of gold-standard functionally validated variants, includes only 213 variants from two genes. Larger functional validation datasets would strengthen confidence in the observed performance differences. While larger high-throughput functional assay datasets exist, such as MaPSy [16], MFASS [2], and Vex-seq [17], these assays focus primarily on exonic and near-exonic variants rather than the deeply intronic variants that represent the primary performance differentiator among the algorithms evaluated here. Additionally, MFASS and Vex-seq are based on minigene reporter systems that lack the native genomic context that SpliceAI and its reimplementations rely on for prediction, potentially limiting the interpretability of any performance comparisons.

Second, for both the Riepe and Barbosa deep intronic datasets, thresholds were optimized per algorithm, which may overestimate real-world performance where a single threshold would be used.

Third, the Extended ClinVar benchmark dataset, while covering a large spectrum of variation, relies on clinical classifications rather than direct functional validation and includes variants with varying levels of evidence quality.

Finally, our study focused exclusively on splice site prediction and did not evaluate other aspects of algorithm performance, such as computational efficiency, runtime, ease of use, or applicability to non-human species.

## Conclusion

In this study, we evaluated the performance of three different deep learning algorithms for splice prediction. Our results indicate that the two open-source reimplementations of SpliceAI successfully replicate the predictive performance of the original algorithm for canonical and near-splice-site variants. CI-SpliceAI and OpenSpliceAI both achieve high balanced accuracy for splice site prediction and represent viable alternatives to the original SpliceAI algorithm, particularly for applications requiring open-source licensing or custom model training.

However, our analysis reveals that algorithm equivalence does not generalize uniformly across all variant classes. On deeply intronic variants, the original SpliceAI outperformed both open-source alternatives. However, both SpliceAI and CI-SpliceAI achieved balanced accuracies at or above 0.89, demonstrating that strong performance on deep intronic variants is attainable with appropriately calibrated thresholds. Nevertheless, all three algorithms required score thresholds an order of magnitude below the standard recommendations to achieve adequate sensitivity, consistent with the findings of Barbosa *et al.* [13].

To fully realize the clinical potential of these algorithms, several priorities emerge for future work. First, comprehensive precomputed score datasets should be generated natively for both GRCh37 and GRCh38 reference assemblies using current transcript annotations, addressing the limitations of existing liftover-derived scores. Second, additional validation using larger functional splice assay datasets would strengthen confidence in the observed performance characteristics, particularly for deeply intronic and non-coding variants. These efforts will facilitate the incorporation of open-source splice prediction tools into routine clinical practice.

## Supporting information

**S1 File. CI-SpliceAI benchmark data.** The data used in the CI-SpliceAI benchmark analysis are available in CI-SpliceAI__Comparison at https://github.com/YStrauch/CI-SpliceAI__Comparison.
(ZIP)

**S2 File. Riepe benchmark data.** The Riepe benchmark data are available in Benchmarking_splice_prediction_tools at https://github.com/cmbi/Benchmarking_splice_prediction_tools.
(ZIP)

**S3 File. ClinVar benchmark dataset.** The ClinVar benchmark dataset is available from bioRxiv at https://www.biorxiv.org/content/10.1101/2025.11.20.689501v1.supplementary-material.
(ZIP)

**S4 File. SPiP benchmark dataset.** The converted SPiP benchmark dataset is available at https://github.com/nlfortier/Analyzing-the-performance-of-deep-learning-splice-prediction-algorithms. The following 15 variants were excluded due to improperly formatted HGVS notation or complex variant types that could not be unambiguously resolved: NM_000059:c.7397C>T, NM_000267:c.888+543T>A, NM_000267:c.2410-9G>A, NM_000267:c.3464A>C, NM_000267:c.5750-332A>G, NM_001042492.2:c.6792C>A, NM_001042492.2:c.6792C>G, NM_001042492.2:c.6792C>T, NM_000267:c.6791C>A, NM_000267:c.7806+2A>C, NM_007294.3:c.177C>A, NM_002611.4:c.1532A>T, NM_000267:c.4773-43151_8315–5891del, NM_007294:c.133_136del, NM_000535:c.24-12_107delinsAAAT.
(ZIP)

**S5 File. Barbosa benchmark data.** The Barbosa benchmark data are available at https://github.com/PedroBarbosa/DeepIntronic_Benchmark.
(ZIP)

## Acknowledgments

We thank our colleagues at Golden Helix for their valuable support and contributions throughout this work. Their insights, feedback, and collaborative spirit were instrumental in the completion of this research.

## Author contributions

**Conceptualization:** Nathan Fortier, Gabe Rudy.

**Formal analysis:** Nathan Fortier.

**Investigation:** Nathan Fortier.

**Methodology:** Nathan Fortier, Gabe Rudy, Andreas Scherer.

**Project administration:** Gabe Rudy, Andreas Scherer.

**Supervision:** Gabe Rudy, Andreas Scherer.

**Writing – original draft:** Nathan Fortier, Gabe Rudy.

**Writing – review & editing:** Gabe Rudy, Andreas Scherer.

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
