## [Decision Letter · Decision Letter 0]

18 Jan 2026

PONE-D-25-68125Analyzing the performance of deep learning splice prediction algorithmsPLOS One

Dear Dr. Fortier,

Thank you for submitting your manuscript to PLOS ONE. After careful consideration, we feel that it has merit but does not fully meet PLOS ONE’s publication criteria as it currently stands. Therefore, we invite you to submit a revised version of the manuscript that addresses the points raised during the review process. Please submit your revised manuscript by Mar 04 2026 11:59PM. If you will need more time than this to complete your revisions, please reply to this message or contact the journal office at plosone@plos.org. Please include the following items when submitting your revised manuscript:

We look forward to receiving your revised manuscript.

Kind regards,

Yang Zhang

Academic Editor

PLOS One

**Journal Requirements:**

“The authors are employees of Golden Helix, Inc., which develops and distributes VarSeq software containing the Legacy Ensemble splice prediction algorithms evaluated in this study. Golden Helix has no commercial relationship with the developers of SpliceAI, OpenSpliceAI, or CI-SpliceAI. The design, conduct, and reporting of this research were not influenced by commercial considerations.”

4. We noted in your submission details that a portion of your manuscript may have been presented or published elsewhere.

“A preprint version of this manuscript has been posted on bioRxiv (doi: 10.1101/2025.11.20.689501). The current manuscript contains substantial revisions and improvements based on feedback received on the preprint. The manuscript is not under consideration at any other peer-reviewed journal.”

“The authors are employees of Golden Helix, Inc., which develops and distributes VarSeq software containing the Legacy Ensemble splice prediction algorithms evaluated in this study. Golden Helix has no commercial relationship with the developers of SpliceAI, OpenSpliceAI, or CI-SpliceAI. The design, conduct, and reporting of this research were not influenced by commercial considerations.”

We note that one or more of the authors are employed by a commercial company: Golden Helix, Inc

**Additional Editor Comments:**

Specifically, please (i) improve benchmark independence by adding more recent, well-curated and/or high-throughput functional splicing datasets and clearly reporting canonical vs non-canonical variant composition, and (ii) revise or justify any pre-filtering steps that may introduce bias. The explanation for OpenSpliceAI’s reduced performance on the Riepe dataset should be supported by direct empirical analysis (e.g., training data/isoform coverage comparison) and, where possible, validated on an expanded functional assay dataset.

Reviewers' comments:

Reviewer's Responses to Questions

**Comments to the Author**

1. Is the manuscript technically sound, and do the data support the conclusions?

Reviewer #1: Yes

Reviewer #2: Yes

2. Has the statistical analysis been performed appropriately and rigorously?

Reviewer #1: Yes

Reviewer #2: Yes

3. Have the authors made all data underlying the findings in their manuscript fully available?

Reviewer #1: Yes

Reviewer #2: Yes

4. Is the manuscript presented in an intelligible fashion and written in standard English?

Reviewer #1: Yes

Reviewer #2: Yes

5. Review Comments to the Author

Reviewer #1: The authors aim to compare CI-SpliceAI, OpenSpliceAI and SpliceAI in an independent manner. This manuscript addresses an important question, as a large proportion of medical genomics is conducted in commercial settings. The authors bring novel findings in this regard.

I have minor revisions to suggest:

The authors base one of their benchmark on the same dataset used by CI-SpliceAI. I have nothing against the dataset from CI-SpliceAI, but independent comparison should try to depart from the original advantageous datasets.

Here are 2 examples of manually curated splice variants lists:

https://pubmed.ncbi.nlm.nih.gov/36273432/, https://pubmed.ncbi.nlm.nih.gov/37878682/

I think these would be more relevant to use instead of the CI-SpliceAI dataset. I think these have a fair good overlap with clinvar variants, but they should be much more clean than clinvar.

Same remark for the multi parallel assays used, published in 2021. I would have expected the authors to use a more recent and more thorough publication like https://pubmed.ncbi.nlm.nih.gov/38129864/

L76 “SpliceAI web interface”, I think the authors are referring to the broad institute web site? if so, please, reformulate.

ll 153-157, I fail to see the logic of this pre-filtering, as it potentially mask some true prediction unseen by the old-school predictors. Indeed, more generally, this pre-filtering introduces some serious bias toward the legacy ensemble. If the authors wish to enrich their dataset in true splicing variants, they could have filtered for the pathogenic variants which are either synonymous or intronic.

Looking manually through some of the pathogenic variants used in ClinVar Benchmark Dataset, I have mostly seen some canonical splice sites. I may be wrong, but I suspect these splice site variants constitute a good deal of the dataset. This might be a slight issue given the goal of the authors to explore the non canonical splice site variants (lines 3-6 of the manuscript). The authors should give this detail to the reader. More generally, this is also interesting to provide for the two other dataset (I mean, the respective proportions of canonical splice variants).

Reviewer #2: This manuscript presents a timely and valuable benchmark of open-source splice prediction tools (OpenSpliceAI and CI-SpliceAI) against the industry-standard SpliceAI. By evaluating these models across multiple datasets, the study addresses a critical need in the field: validating permissively licensed alternatives that can be integrated into clinical pipelines. The confirmation that these open-source models generally reproduce the predictive power of the original SpliceAI is an important resource for the community. However, to fully establish these tools as reliable alternatives for clinical diagnostics, further rigorous investigation into specific performance discrepancies, particularly regarding functional validation, is necessary.

Major Comments

1. The authors currently attribute OpenSpliceAI's significantly lower performance on the Riepe dataset to "gene-specific splicing patterns in ABCA4 and MYBPC3 that are not fully captured by OpenSpliceAI's training data". This explanation is currently speculative and, as a defense for a tool intended for genome-wide application, inadequate without empirical backing.

It remains unclear whether this poor performance stems from the architecture/methodology or simply the training data source (RefSeq MANE vs. GENCODE). Since the splice sites used for the original SpliceAI training are accessible in its repository, it would be highly beneficial to directly compare the splice site overlap between the SpliceAI and OpenSpliceAI training sets. Specifically, the authors should verify if the isoforms or intronic regions relevant to these ABCA4 and MYBPC3 variants were excluded from the RefSeq MANE training set. Confirming whether these sites were "seen" by one model and not the other would definitively clarify if the failure is a data coverage issue.

2. Expanding Functional Assay Benchmarking While the inclusion of the Riepe dataset is a valuable starting point, its small sample size (N=213) and restriction to only two genes limit the ability to draw broad conclusions about the functional reliability of OpenSpliceAI compared to the original SpliceAI. The observed performance gap on this dataset is concerning, and N=213 is insufficient to determine if this is a systematic issue with "cryptic" splicing or an artifact of those specific genes.

To robustly validate the tools, I strongly recommend expanding the benchmark to include larger, high-throughput functional datasets. Incorporating even a subset of the following resources would significantly strengthen the manuscript’s conclusions:

* MFASS (Multiplexed Functional Assay of Splicing using Sort-seq): (Cheung et al., 2019)

* MaPSy (Massively Parallel Splicing Assay): (Soemedi et al., 2017)

* Vex-seq (Variant Exon Sequencing): (Adamson et al., 2018)

Expanding the evaluation to include one or more of these high-throughput datasets would move the study beyond a simple reproduction effort and establish it as a definitive benchmark for the clinical community. It would also help clarify whether the current performance discrepancies are limited gene-specific artifacts or systematic differences in how the open-source models handle cryptic splicing.

6. PLOS authors have the option to publish the peer review history of their article (what does this mean?). If published, this will include your full peer review and any attached files.

Reviewer #1: **Yes:** Jean-Madeleine de Sainte Agathe

Reviewer #2: No

---

## [Author Response · Author response to Decision Letter 1]

9 Mar 2026

We thank the editor and reviewers for their thoughtful and constructive comments. We have carefully considered each point raised and have revised the manuscript accordingly. Below we provide a point-by-point response to each comment.

Editor Comments

Comment 1: Improve benchmark independence by adding more recent, well-curated and/or high-throughput functional splicing datasets and clearly reporting canonical vs non-canonical variant composition.

Response: We have incorporated three additional benchmark datasets into the study: (1) the SPiP dataset (Leman et al., 2022), comprising 99,601 variants with experimentally validated splicing outcomes across 227 genes; (2) the Barbosa deep intronic benchmark (Barbosa et al., 2023), comprising 242 manually curated pathogenic deep intronic variants; and (3) the Barbosa ClinVar benchmark (Barbosa et al., 2023), comprising 53,600 intronic variants from ClinVar. We have also added explicit reporting of the proportion of variants altering the canonical splice site dinucleotides for each dataset. For the Barbosa deep intronic benchmark, canonical splice site variants are excluded by design, as all variants are required to be located more than 10 bp from the nearest annotated splice site.

Comment 2: Revise or justify any pre-filtering steps that may introduce bias.

Response: We have revised the Methods section to more explicitly acknowledge the potential bias introduced by the Legacy Ensemble pre-filtering step in the Extended ClinVar benchmark. Specifically, we now note that this filtering strategy may systematically exclude variants detectable only by deep learning models but not by any Legacy Ensemble algorithm, potentially conservatively underestimating the specificity advantage of deep learning approaches. We further note that this limitation is partially mitigated by the inclusion of the Barbosa deep intronic benchmark, which evaluates algorithm performance on deeply intronic variants without any pre-filtering based on legacy tool predictions.

Comment 3: The explanation for OpenSpliceAI's reduced performance on the Riepe dataset should be supported by direct empirical analysis and, where possible, validated on an expanded functional assay dataset.

Response: We have replaced the speculative explanation with a direct empirical comparison of the GENCODE v24 annotation used to train SpliceAI and the RefSeq MANE Select annotation used to train OpenSpliceAI at both the ABCA4 and MYBPC3 loci. This analysis revealed two key findings. First, ABCA4 resides on a held-out chromosome under the data split applied by both models, meaning neither was trained on ABCA4 splice sites, and any performance difference at this locus reflects generalization capacity rather than training data coverage. Second, while GENCODE v24 contributed three protein-coding isoforms for MYBPC3 encompassing 36 unique introns compared to the single MANE Select transcript with 34 introns, none of the Riepe intronic and splice-region variants in the MYBPC3 dataset fell within the GENCODE-exclusive introns. These findings rule out annotation coverage as an explanation for the observed performance gap. We further note that the convergence of this finding with the results on the Barbosa deep intronic benchmark, where OpenSpliceAI also underperformed relative to SpliceAI, suggests a more general difficulty with deeply intronic variant detection rather than a gene-specific effect.

Reviewer 1

Comment 1: The authors base one of their benchmarks on the same dataset used by CI-SpliceAI. Independent comparison should try to depart from the original advantageous datasets. Two examples of manually curated splice variant lists were suggested: https://pubmed.ncbi.nlm.nih.gov/36273432/ and https://pubmed.ncbi.nlm.nih.gov/37878682/.

Response: We thank the reviewer for this suggestion. We note that the two publications referenced correspond to the SPiP dataset (Leman et al., 2022) and the Barbosa benchmark (Barbosa et al., 2023), both of which have been incorporated into the revised manuscript as described above. We have retained the CI-SpliceAI benchmark dataset as it provides a useful point of comparison with the performance reported in the original CI-SpliceAI publication, but the addition of these independent datasets substantially strengthens the benchmark independence of our study.

Comment 2: The SpliceAI web interface reference should be reformulated.

Response: We have revised the text to explicitly identify this as the Broad Institute SpliceAI Lookup web interface and have included the URL directly in the manuscript text.

Comment 3: The pre-filtering step potentially masks true predictions unseen by legacy predictors and introduces bias toward the legacy ensemble.

Response: We have revised the Methods section to explicitly acknowledge this limitation, as described in our response to Editor Comment 2 above.

Comment 4: The proportion of canonical splice site variants should be reported for all datasets.

Response: We have added explicit reporting of canonical splice site dinucleotide-altering variant proportions for all datasets, as described in our response to Editor Comment 1 above.

Reviewer 2

Comment 1: The explanation for OpenSpliceAI's reduced performance on the Riepe dataset is speculative and inadequate without empirical backing.

Response: We have addressed this with direct empirical analysis as described in our response to Editor Comment 3 above.

Comment 2: The functional assay benchmarking should be expanded to include larger high-throughput datasets such as MFASS, MaPSy, and Vex-seq.

Response: We thank the reviewer for these suggestions. We have considered each of these resources carefully. However, these datasets present some limitations for our specific benchmarking goals. MaPSy (Soemedi et al., 2017) and MFASS (Chong et al., 2019) focus primarily on exonic variants, and MFASS and Vex-seq (Adamson et al., 2018) are based on minigene reporter systems that lack the native genomic context that SpliceAI and its reimplementations rely on for prediction, potentially limiting the interpretability of performance comparisons. Most importantly, all three datasets focus on exonic and near-exonic variants rather than the deeply intronic variants that represent the primary performance differentiator among the algorithms evaluated here. We have acknowledged these resources and their limitations in the Study Limitations section of the revised manuscript. The addition of the SPiP and Barbosa datasets, which include experimentally validated splicing outcomes across a broad range of variant types and genomic contexts, substantially expands the functional benchmarking scope of the study beyond the original Riepe dataset.

We believe these revisions comprehensively address all comments raised by the editor and reviewers and have substantially strengthened the manuscript. We thank the reviewers again for their constructive engagement with our work.

---

## [Decision Letter · Decision Letter 1]

22 Apr 2026

Analyzing the performance of deep learning splice prediction algorithms

PONE-D-25-68125R1

Dear Dr. Fortier,

We’re pleased to inform you that your manuscript has been judged scientifically suitable for publication and will be formally accepted for publication once it meets all outstanding technical requirements.

Kind regards,

Yang Zhang

Academic Editor

PLOS One

Additional Editor Comments (optional):

Reviewers' comments:

Reviewer's Responses to Questions

**Comments to the Author**

1. If the authors have adequately addressed your comments raised in a previous round of review and you feel that this manuscript is now acceptable for publication, you may indicate that here to bypass the “Comments to the Author” section, enter your conflict of interest statement in the “Confidential to Editor” section, and submit your "Accept" recommendation.

Reviewer #1: All comments have been addressed

Reviewer #2: All comments have been addressed

2. Is the manuscript technically sound, and do the data support the conclusions?

Reviewer #1: Yes

Reviewer #2: Yes

3. Has the statistical analysis been performed appropriately and rigorously?

Reviewer #1: Yes

Reviewer #2: Yes

4. Have the authors made all data underlying the findings in their manuscript fully available?

Reviewer #1: Yes

Reviewer #2: Yes

5. Is the manuscript presented in an intelligible fashion and written in standard English?

Reviewer #1: Yes

Reviewer #2: Yes

6. Review Comments to the Author

Reviewer #1: I thank the authors for their revisions, which improved the quality of this manusript. They have better explained and clarify several points.

Reviewer #2: I would like to commend the authors for their thorough and thoughtful revisions to the manuscript. The revised version successfully addresses the primary concerns raised during the initial review.

Most notably, the incorporation of the three independent, external benchmark datasets—the SPiP dataset, the Barbosa deep intronic benchmark, and the Barbosa ClinVar benchmark—substantially strengthens the study's findings. This approach to external benchmarking provides a much more robust, reliable, and unbiased evaluation of the splice prediction algorithms.

The authors have fully satisfied my previous concerns regarding benchmark independence. The manuscript is much improved, and the conclusions are now well-supported by independent data.

7. PLOS authors have the option to publish the peer review history of their article (what does this mean?). If published, this will include your full peer review and any attached files.

Reviewer #1: **Yes:** Jean-Madeleine de Sainte Agathe

Reviewer #2: No

---

## [Editor Report · Acceptance letter]

PONE-D-25-68125R1

PLOS One

Dear Dr. Fortier,

I'm pleased to inform you that your manuscript has been deemed suitable for publication in PLOS One. Congratulations! Your manuscript is now being handed over to our production team.

Kind regards,

on behalf of

Dr. Yang Zhang

Academic Editor

PLOS One